# Predictors of self-reported practice in ventilator-associated pneumonia (VAP) prevention among critical care nurses in Sarawak public hospitals

Jia Hui Tan (ID), Chong Chin Che (ID)*, Li Yoong Tang, Mei Chan Chong

Department of Nursing Science, Faculty of Medicine, Universiti of Malaya, Kuala Lumpur, Malaysia

* chechongchin@um.edu.my

## Abstract

### Background and objective

Ventilator-associated pneumonia (VAP), a leading cause of ICU mortality, remains prevalent in Southeast Asia, with limited data on critical care nurses' knowledge and prevention practices in Malaysia. The purpose of this study was to assess knowledge, self-reported practices, barriers, and predictors of VAP prevention among critical care nurses in Sarawak, Malaysia.

### Methods

This cross-sectional study was conducted from July to August 2023 at four public hospitals in Sarawak, Malaysia. Universal sampling was used to recruit nurses from various critical care units managing patients requiring ventilator support. A self-administered questionnaire, consisting of four sections, was employed to gather background information from nurses, assess their knowledge, self-reported practices, and identify barriers related to VAP prevention.

### Results

A total of 298 critical care nurses participated in the study. Of these, 66.8% demonstrated poor knowledge of ventilator-associated pneumonia (VAP) prevention; however, self-reported practices with VAP prevention were significantly high at 76.5%. A Pearson's correlation test revealed a significant association between the nurses' knowledge and their self-reported practices related to VAP prevention ($p < 0.001$). Additionally, multiple regression analyses identified several significant predictors of critical care nurses' self-reported practices in VAP prevention, including their level of knowledge, type of unit, number of official beds, and sociodemographic factors ($p < 0.05$). While knowledge positively influenced self-reported practices, its impact was relatively minor compared to sociodemographic factors. Barriers to VAP prevention included nursing staff shortages, forgetfulness, and lack of written protocols.

**Data availability statement:** All relevant data are within the manuscript and its Supporting Information files.

**Funding:** The author(s) received no specific funding for this work.

**Competing interests:** The authors have declared that no competing interests exist.

## Conclusions

The prevention of ventilator-associated pneumonia (VAP) is a multidisciplinary challenge, emphasizing the crucial role of critical care nurses. The findings from this study underscore the necessity for updated, evidence-based interventions that target knowledge gaps, self-reported practices, and barriers to effective VAP prevention.

## Introduction

Mechanical ventilation is used to offer ventilator support regularly to patients in intensive care units (ICUs). However, mechanical ventilation is associated with a high incidence of adverse outcomes, such as mortality, prolonged ventilation, and extended hospital stays, increasing the financial burdens on patients and creating a high demand for healthcare resources [1]. Ventilator-associated pneumonia (VAP) is one of the most common hospital-acquired infection (HAI) occurring in ICUs and is a major cause of mortality among patients admitted to such units [2]. Recent studies have reported a worrisome prevalence of VAP in various Southeast Asian countries, indicating a significant rise in mortality rates from 16.2% to 74.1% and VAP incidence ranging from 2.13 to 116 per 1,000 patient-days from the year 2000 and 2020 [3]. In Malaysia, a tertiary centre reported a VAP incidence density of 7.1 per 1,000 ventilator-days [4].

Many guidelines and protocols have been proposed, but there is still a gap in the translation of such guidelines and protocols into bedside among critical care nurses [5]. Several factors that may account for this gap in care delivery include unfamiliarity with recommendations, content, time, resources, education, and training in VAP prevention, as well as protocol compliance [5]. Nurses who provide around-the-clock bedside care to mechanically ventilated patients play a crucial role in implementing non-pharmacological prevention measures that are directly related to lower VAP incidence rates [6]. Nurses should, therefore, be fully informed and apply evidence-based practices in their daily practice to minimize the incidence of VAP among ICU patient.

Concerns have been raised about the ability of nurses in critical care to prevent VAP. Zeb et al. reported that critical care nurses demonstrated a poor understanding of VAP prevention measures [7]. Similarly, a systematic review by Al-Mugheed et al. included 23 studies, of which 14 reported a poor level of awareness regarding VAP knowledge among nurses [8]. The general results determined an inadequate level of compliance with the VAP prevention measures among institutions. For effect, such knowledge gaps could thwart the successful delivery of VAP preventive practices and the translation of interventions into practice. Notably, critical care nurses caring for ventilated patients be knowledgeable of the guideline-recommended evidence-based guideline practices for VAP prevention. Indeed, nurses with high-level understanding of VAP prevention show high levels of engagement in practices that aim to reduce the incidence of VAP.

The fundamental principles of evidence-based care for preventing VAP must align with compliance measures to translate the essential elements into effective practice would thus need to conform to the basic elements of evidence-based care to prevent VAP. The research on nurses' perceptions of barriers for preventing VAP has unraveled a heterogeneous set of contributing factors, including suboptimal working conditions, poor organizational culture, deficiencies in educational resources and policy frameworks, and poor protocol guidelines, in addition to poor staffing levels [9].

The main aim of this research was to assess the knowledge and self-reported practices about prevention strategies for VAP among critical care nurses, various knowledge and self-reported practices variations based on sociodemographic characteristics, and the predictors of their self-reported practices on the prevention of VAP. The study also established barriers to compliance with prevention protocol. To date, there is inadequate information comprehensively describing the knowledge, practice and barriers to VAP prevention among nurses working in critical care areas in Malaysia.

## Methods

### Study design

An analytic cross-sectional study design was adopted, adhering to the Strengthening the Reporting of Observational Studies in Epidemiology (STROBE) guidelines.

**Setting and samples.** Self-administered surveys were distributed to all registered nurses working in four public hospitals in Sarawak. A total of 298 nurses from various critical care settings participated in the study. A multistage sampling method was employed to select the study settings, while universal sampling was used to recruit nurses from both adult and pediatric critical care units who were caring for ventilated patients. The inclusion criteria were as follows: (1) The participant had to be a registered nurse and worked for more than three months in intensive care units, including high dependency units, and (2) the participants must have minimal qualifications in Diploma in Nursing. All registered nurses under probation or mentoring, on study leave, maternity leave, or long sick leave were excluded from the study.

**Data collection.** The data collection took place from July 3rd, 2023, to August 30th, 2023 during which questionnaires were distributed in physical form to all registered nurses in critical care units within the selected settings. Responses with incomplete data or entry errors were excluded. Incomplete data were defined as any questionnaire with missing responses to demographic or knowledge and practice items, as all items were considered essential for analysis. Only fully completed questionnaires were retained. A total of 12 incomplete questionnaires were excluded, leaving 298 valid responses for analysis. Additionally, thirty nurses were excluded from the analysis as their responses were used for a pilot study.

### Measurements

**Demographic characteristics.** A self-administered questionnaire was used, and the items covering demographic data and ICU characteristics: age, gender, education level, years of nursing experience, years of ICU experience, advanced specialty training in intensive care nursing (locally termed as post-basic qualification in ICU), type of unit, number of official beds, nurse-patient ratio for ventilated patients, job position, working hours per week, employment status, participation in in-service education and training on VAP prevention, as well as the source of education and training on VAP prevention strategies.

**Knowledge of VAP prevention strategies.** To evaluate the nurses' knowledge, an English version of the self-reported tool was adopted [10] in this study. The scale contains fourteen dichotomous questions related to knowledge of VAP prevention strategies. All items are rated with 1 point for correct answers, 0 points for incorrect answers or "Don't know". Total scores ranged from 0 to 14, with higher values indicating greater knowledge. For analysis, the total score was converted into a percentage. In line with Al-Sayaghi's cut-off point, knowledge levels were categorized into three groups: Good (70–100%), Acceptable (60–69%), and Poor (≤60%) [10]. The total scale had a Cronbach's α of.64, the Content Validity Index (CVI) was.89, and this scale was valid and reliable.

**Self-reported practices towards VAP prevention.** The compliance scale used in this study was also the English version compiled by Al-Sayaghi, which was adopted to assess nurses' compliance with nursing-relevant VAP prevention practices [11]. All items are evaluated using a five-point Likert scale (never, sometimes, always). A higher total score reflected greater compliance with guidelines. The score was then converted to a percentage relative to the total score. In line with Al-Sayaghi's classification [11], the results were grouped into three categories of high compliance (75–100%), acceptable compliance (50–75%), and unsafe practice (<50%). The Cronbach's α score of the total scale was.70, CVI was.94, and this scale was valid and reliable.

**Barriers towards VAP prevention.** The barriers scale used in this study was also adapted from Al-Sayaghi [12] to determine the possible barriers preventing adherence to VAP prevention guidelines. All the items are evaluated using a 3-point Likert scale (agree, neither agree nor disagree, disagree). The total count of those who either agreed or disagreed in the fourth section was arranged from the highest to the lowest agreement. The Cronbach's α score of the total scale was.80, and the CVI was.94, and this scale was valid and reliable.

### Ethical considerations

This study received ethical approved from Medical Research Ethics Committee (MREC) under the National Medical Research Register (NMRR) Malaysia (Reference no: NMRR ID-23–00596-MTN (IIR)). Participants were assured of data confidentiality and anonymity and their participation was entirely voluntary.

### Data analysis

Data analysis for the study was conducted using SPSS version 26.0. Results were presented as frequency, percentage, and mean±standard deviation (SD). The mean values between two independent groups were compared using Independent T-test, while comparisons among multiple groups were made using one-way ANOVA. For data that were not normally distributed or violated the assumption of homogeneity of variance, Welch T test and Welch ANOVA were employed. Pearson's correlation was used to assess the relationships between knowledge and self-reported practice.

Hierarchical multiple regression analysis was used to identify predictors of self-reported practice in VAP prevention. In Model 1, organizational and workload-related variables (type of unit, number of official beds, and nurse–patient ratio for ventilated patients) were entered. In Model 2, knowledge score was added to see if it explained additional variance beyond these organizational factors. Statistical significance was set at $p < .05$.

### Results

Out of 454 eligible critical care nurses, the minimum required sample size was 213, as calculated using Yamane's formula. After adjustment for a 20% attrition rate, the final target sample size was 267. The researchers successfully obtained 298 completed responses, which exceeded the required sample size and represented a response rate of 65.6%. This higher-than-expected response rate strengthens the reliability and generalizability of the study findings [12].

Of the 298 critical care nurses' respondents, 285 (95.6%) were women and 13 (4.4%) were men. Participants' ages ranged from 20 to 40 years, with the majority (52.0%) having 10 years or less of nursing experience. Most (96.0%) had a diploma, and 48.7% had 6–7 years of experience in the Intensive Care Unit (ICU). Additionally, 79.2% did not possess advanced specialty training in intensive care nursing (locally termed as post-basic qualification in ICU). The participants represented diverse critical care settings, with 42.6% working in a General Intensive Care Unit. Regarding facility size, 44.6% were in settings with fewer than 10 beds and 88.6% cared for two ventilated patients. All participants were nurses. The majority (74.8%) were bedside clinical staff nurses (entry-level registered nurses), while 20.5% were senior registered nurses with 13–15 years of service, promoted through structured time-based pathways, who frequently take on supervisory and mentoring roles in addition to their clinical duties. A smaller proportion (4.7%) were nurse managers (ward or unit leaders with broader administrative responsibilities). Majority worked fewer than 45 hours per week (58.1%), and

98.0% were permanent staff members. Furthermore, the majority had received education and training on VAP prevention (87.6%), with 69.8% of that training obtained through in-service education. Characteristics of critical care nurses described in "Table 1".

### Nurses' knowledge of VAP prevention

In general, the knowledge score about VAP prevention was low 7.40±2.24 as shown in "Table 2". The sequence of individual items from highest to lowest scores in the knowledge questionnaire was as follows: regular comprehensive oral care (82.0%), frequency of the ventilator circuit changes (75.0%), type of airway humidifier (75.0%) and oral versus nasal route for endotracheal intubation (71.0%) (S1 Table). Significant variables associated with the nurses' knowledge about the prevention of VAP in this group were age, years of nursing experience, ICU nursing experience, unit type, number of official beds and job position.

### Self-reported practices towards VAP prevention

The general score of self-reported practices to prevent VAP was high: 27.98±3.04, indicating a high level of compliance, as shown in "Table 3". In rank order from most frequent to least, the items were as follows: washing hands after any approach to a patient, 99.0%, washing hands before any approach to a patient, 98.7% using sterile gloves when open suction is necessary, 96.6%; and using the heat and moisture exchanger humidifier, 88.6% (S2 Table). The variables that are significantly associated with compliance by critical care nurses with the VAP prevention guidelines are the type of units, number of official beds, and the nurse-to-patient ratio for ventilated patients.

### Correlation between knowledge and self-reported practices towards VAP prevention

Pearson's correlation analysis was used to examine the relationship between knowledge and self-reported practice towards VAP prevention. There was a statistically significant positive correlation between the two variables. Precisely, knowledge was positively, although weakly associated with the critical care nurses' practice concerning VAP prevention: $r = .204$, $p < .001$ as shown in "Table 3". From this study, the researchers identified a correlation between knowledge and self-reported practices. The correlation showed a weak positive relationship of these two factors. This follows through from the statistically significant implication of clinical practice by poor knowledge and also positive self-reported compliance with the practice of VAP prevention by the critical care nurses [13].

### Barriers towards VAP prevention

Barriers to the prevention of VAP by nurses have been ranked from highest to lowest based on their scores and level of agreement. The highest-rated barriers were shortage of staff in the unit, forgetting to perform certain evidence-based procedures that may elevate the risk for VAP, and lack of written protocol on the management of ventilator-associated pneumonia. The items rated highest were those related to potential barriers in adhering to the guidelines for the prevention of VAP. Only variables of post basic training in ICU, type of units, number of official beds, and participation in-service education and training showed significantly associated with barriers faced by critical care nurses in preventing VAP (S3 Table).

### Predictors of critical care nurses' self-reported practices in VAP prevention

Hierarchical multiple regression analysis further revealed that knowledge was positively related to practices. Sociodemographic variables of type of units, number of beds, and nurse-to-patient ratio for ventilated patients accounted for 12.9% of the variance in self-reported practices in Model 1 ($R^2 = .129$, $r = .36$, $F = 14.55$, $p < .001$) as shown in "Table 4". As seen, Model 2 canvases VAP prevention knowledge to comfortably increase the explanatory power of this model, accounting for 14.1% of the variance in self-reported practice ($R^2 = .141$, $r = .37$, $F = 12.03$, $p < .001$. It added another 1.2% to the variance.

**Table 1. Sociodemographic characteristics of critical care nurses (N = 298).**

| Characteristics | Frequency (n) | Percentage (%) |
|---|---|---|
| **Gender** | | |
| Male | 13 | 4.4 |
| Female | 285 | 95.6 |
| **Age (in years)** | | |
| 20-30 | 55 | 18.5 |
| 31-40 | 185 | 62.1 |
| >40 | 58 | 19.4 |
| **Education level** | | |
| Diploma | 286 | 96 |
| Bachelor's Degree | 12 | 4 |
| **Total nursing experience (years)** | | |
| ≤10 | 155 | 52 |
| >10 | 143 | 48 |
| **ICU nursing experience (years)** | | |
| ≤5 | 75 | 25.2 |
| 6 −10 | 145 | 48.7 |
| >10 | 78 | 26.2 |
| **Completed advanced certification in intensive care nursing** | | |
| Yes | 62 | 20.8 |
| No | 236 | 79.2 |
| **Type of units** | | |
| General Intensive Care Unit | 127 | 42.6 |
| Infectious Disease Intensive Care Unit | 39 | 13.1 |
| Pead Intensive Care Unit | 42 | 14.1 |
| Neuro High Dependency Unit | 34 | 11.4 |
| Medical High Dependency Unit | 56 | 18.8 |
| **Number of official beds** | | |
| ≤10 | 133 | 44.6 |
| 11 - 15 | 62 | 20.8 |
| 16 - 20 | 67 | 22.5 |
| >20 | 36 | 12.1 |
| **Nurse patient ratio for ventilated patients** | | |
| 1:1 | 34 | 11.4 |
| 1:2 | 264 | 88.6 |
| **Job position** | | |
| Clinical Staff Nurse | 223 | 74.8 |
| Senior clinical Staff Nurse | 54 | 18.1 |
| Clinical nurse manager | 21 | 7.0 |
| **Working hours/week** | | |
| ≤45 | 173 | 58.1 |
| >45 | 125 | 41.9 |
| **Employee status** | | |
| Permanent | 292 | 98.0 |
| Contract | 6 | 2.0 |
| **Participation in-service education& training on VAP prevention** | | |
| Yes | 261 | 87.6 |
| No | 37 | 12.4 |

*(Continued)*

**Table 1.** (Continued)

| Characteristics | Frequency (n) | Percentage (%) |
|---|---|---|
| **Sources of education & training on VAP prevention** | | |
| In-service education | 208 | 69.8 |
| Nursing schools | 63 | 21.1 |
| Web internet resources | 46 | 15.1 |

**Table 2.** Critical care nurses' knowledge level of ventilator-associated pneumonia (VAP) prevention.

| Variables | Range of score | n (%) | Mean (SD) |
|---|---|---|---|
| **Knowledge level** | | | |
| Good knowledge | 9.8-14 (≥70.0-100%) | 48 (16.1) | 7.40±2.24 |
| Acceptable knowledge | 8.4-9.7(60.0-69.0%) | 51 (17.1) | |
| Poor knowledge | 0-8.3 (0-60.0%) | 199 (66.8) | |

*Note:* n: frequency

%: percentage

SD: standard deviation

**Table 3.** Critical care nurses' self-reported practices level and correlation with knowledge of ventilator-associated pneumonia (VAP) prevention (N = 298).

| Variables | Range of score | n (%) | Mean (SD) | 1 | 2 |
|---|---|---|---|---|---|
| **Practice level** | | | | | |
| High compliance | ≥23-32 (≥ 75.0-100%) | 228 (76.5) | 27.98±3.04 | | |
| Acceptable compliance | 16-23 (50.0-75.0%) | 70 (23.5) | | | |
| Unsafe practice | <16 (<50.0%) | 0 | | | |
| **Correlation between variables** | | | | | |
| 1. Knowledge | – | 298 | 7.40±2.24 | 1 | .204* |
| 2. Practice | – | 298 | 27.98±3.04 | .204* | 1 |

*Note:* n: frequency

%: percentage

SD: standard deviation

* $p < .001$

Knowledge, although positive effect (B = .156, $p < .001$), means it improves the ability to explain variance in the dependent variable. It had a partly effect when compared with sociodemographic factors ($\Delta R^2 = .012$). while knowledge also has a significant role, the sets of sociodemographic characteristics have more significant predictive power on self-reported practice regarding the prevention of VAP.

## Discussion

The overall knowledge of critical care nurses regarding the prevention of VAP was found to be decidedly low, below the acceptable threshold ≤ 60.0%, and was consistent with several other studies [11,13–15]. The lowest accuracy rate, as reported by the nurses, was for statements on the frequency of humidifier heat and moisture exchanger changes at 98.0% accuracy. Although recommendations generally suggest that the frequency of change for heat and moisture exchangers (HMEs) be weekly or as clinically indicated, practices related to HME replacement and change frequency vary widely. In General ICUs, for example, HMEs are changed every shift; in other critical care units, they are replaced every 24 hours

**Table 4. Predictive Model: Hierarchical Multiple Regression Analysis for Self-Reported Practices of Ventilator-Associated Pneumonia (VAP) Prevention.**

| Variables | Practice towards VAP prevention | | | |
| --- | --- | --- | --- | --- |
| | Model 1 | | Model 2 | |
| | *B* | *β* | *B* | *β* |
| Constant | 27.28** | 1.32 | 25.98** | 1.46 |
| Type of units | −.836** | .132 | −.773** | −.399 |
| Number of official beds | .374* | .197 | .407* | .144 |
| Nurse patient ratio for ventilated patients | 1.08 | .653 | 1.04* | .109 |
| Knowledge | | | .156 | .115 |
| R² | .129 | | .141 | |
| F | 14.55** | | 12.03** | |
| ΔR² | .120 | | .129 | |

a. *Note*: B = regression coefficient; *β* = beta coefficient; △R² = adjusted R2; **p < .001; *p < .05

b. Predictor: Type of units, number of official beds, and nurse patient ratio for ventilated patients (continuous variable),

c. Dependent variable: self-reported practices towards VAP prevention

ICU: intensive care units; VAP: ventilator-associated pneumonia

or as clinically indicated, such as if clogged. Whereas most manufacturers recommend replacing HMEs every 24 hours, there is debate over this recommendation; for instance, it has been suggested that alternative intervals could be 48 hours, 96 hours, or even on a weekly basis [16,17]. Logically, HMEs should be checked for signs of resistance every 24 hours and replaced as necessary, which may be required every few days, though the time frame does vary significantly [17]. 77.0% of nurses also scored poorly in identifying different types of endotracheal tubes, similar to those in previous studies [18–21]. Recommendations include the use of subglottic secretion drainage (SSD-ETTs) over the traditional endotracheal tubes. Although financial constraints favour the use of standard ETTs, guidelines on the prevention of VAP recommend SSD-ETTs due to their efficacy in reducing VAP rates and shorter ICU stays without increased adverse events or mortality [22]. However, SSD-ETTs have low and underutilized adoption in ICUs [23].

The majority of the nurses expressed a high 76.2% and an acceptable 23.5% self-reported practice regarding VAP prevention, and aligning with earlier studies [24–26]. One reason for this is the inclusion of the bundle for the prevention of VAP prevention in the undergraduate nursing curriculum [24]. Consequently, subglottic suctioning through an additional lumen in ETTs contributed to less likelihood in 49.7% of the nurses. In regard to these observations, a study by Al-Sayaghi [24], also revealed several shortcomings related to coursework and curricula in undergraduate nursing. Till recently, there were only a few ETTs available with an additional lumen, and compliance rates for performing subglottic secretion suctioning ranged from 5.8% to 32.6% between 2010 and 2016 [27]. Indeed, Jahansefat et al. [28] reported that some settings recorded no usage of interventions related to subglottic secretion. The general low compliance score in this item could thus be due to unfamiliarity with types of ETTs designed for the facilitation of subglottic secretion drainage. Standard endotracheal tubes were widely used across all units, where the secretions were ventilated and suctioned through one tube. This obviously reduces the use of the subglottic drainage tube.

Nurses were even reluctant to perform activities regarding the use of chlorhexidine solution for oral care 39.3%. it was probably due to prevailing ICU practice and customs observed in general ICUs, which have been discontinued based on recommendations by intensivists, which are in line with updated evidence-based practice [28]. There were similar trends in pediatric units where chlorhexidine was not suitable and for which there is some evidence that it does not reduce VAP in critically ill children [29]. In Neuro High Dependency Unit's (HDU), chlorohexidine gluconate mouthwash was used

for oral care. The notable point here is that no standardized prevention measures were made for use by nurses in their respective settings. The Center for Disease Control and Prevention (CDC) does not recommend the routine oral administration of chlorhexidine mouthwash while awaiting further data; conflicting recommendations are made between the most recent evidence and institutional policies [29–31]. Recent evidences recommended a worldwide cessation of chlorhexidine mouthwash use [32,33]. Ricard & Lisboa [34] recommended that more evidence was needed before its use in ventilated patients was stopped. Such inconsistencies in the literature may have influenced nurses' responses in this study. In particular, the significant barriers to the implementation of guidelines for VAP prevention include a shortage of nursing staff, which is consistent the findings from other studies [9,15,24,35–38]. Poor staffing invariably results in heavy workloads for the nurses, especially for those working in the ICUs. This indirectly affects the quality of care accorded to the patients and compromise patient safety in general [38].

Such findings indicate a gap between the evidence-based guidelines on VAP prevention and the actual knowledge and practices among CCNs, suggesting possible risks to patient safety, quality of care and clinical outcomes. For instance, while 98.7% of the nurses reported self-reported adherence to trusted practice in the prevention of VAP, 66.8% of the nurses had inadequate knowledge of VAP prevention. This contrast suggests that the high compliance reported may not fully reflect actual practice, but could be influenced by reporting tendencies, underscoring the need for more objective assessments. So far, only one study has demonstrated the existence of such findings, and few studies have addressed this issue yet [39]. This disparity thus indicates that knowledge gaps may be translated into inappropriate practice in preventive measures of VAP, leading to higher rates of VAP. Future professional development through targeted program and continued education would play a role in enhancing CCNs' competencies, making care practice standardize, thereby improving patient's outcome by reducing the incidence of VAP [38–43]. Moreover, it needs to be considered whether stress and workload factors affect nurses' compliance with the VAP protocols. The ability to address these factors could provide a practical way of motivating CCNs to follow these guidelines consistently.

Hierarchical analysis resulted in only a significant negative association between unit type and self-reported practices toward VAP prevention. It may be perceived that certain unit types may encounter struggles in maintaining effective VAP prevention measures compared to others. This observation is supported by the study conducted by Al-Sayaghi [24]. He highlighted that the type of ICU influences adherence to VAP guidelines. These findings presented differences in VAP prevention practices across different clinical settings. The number of beds also has a positive and significant impact on self-reported practice, as supported by several studies [24,43]. The units with higher bed counts have a positive and significant impact on self-reported practices in the prevention of VAP. This would imply that larger units with more beds may have more resources, staffing, or structured protocols that contribute to better adherence to guidelines on VAP. The nurse-to-patient ratio for ventilated patients shows a positive association with self-reported practice in both models, although at a non-statistically significant level. Higher ratios may positively impact practices related to VAP prevention and thus reinforce notions of adequate staffing being imperative for the delivery of quality care. This finding is in agreement with the study conducted by Aloush [4], which had similar results. From the analysis, it also emerged that knowledge has a slight positive influence on self-reported practices toward VAP prevention, though minimal in effect size and not statistical significance. This finding agrees with that by Darawad [25], who reported a statistically significant positive correlation between VAP guideline adherence and total VAP care knowledge (r = 0.175, $p > .05$). The nonsignificant influence in this study implies that knowledge alone may not be strong enough to ensure meaningful changes in self-reported practice; it nonetheless plays an assuring role in guiding adherence to established guidelines. Knowledge improvement may be, therefore the cornerstone for improving VAP prevention practices, complemented by other factors, including sufficient staffing, practical training, and institutional support in effectively improving VAP prevention practices.

These findings support the updating and standardization of guidelines on VAP prevention in various healthcare settings, due to staffing shortages and workloads affecting adherence. Encourage the nurses to participate in quality improvement projects, have regular professional growth and training, and provide a supportive culture for the nurses to apply the VAP prevention strategies.

 

For nursing education, extensive training on VAP pathophysiology and prevention, simulation-based learning and emphasis on communication, teamwork, and safety are all critical [41]. Nursing research shall study the obstacles to VAP prevention by investigating the workload of the staff against the availability of resources for infection control. Consequently, it is with the help of qualitative research that an emphasis on such issues will pay dividends in policy changes and result in better outcomes for the patients.

## Limitations

Several limitations are identified in this present study. The findings of this study, which investigated the knowledge of CCN's, their self-reported practice, and the barriers to VAP prevention, have limited generalizability because of the cross-sectional design and the focus on ICUs in tertiary hospitals within the state of Sarawak. Future studies should cover CCNs working in various ICUs in different types of hospitals and states of Malaysia to enhance the generalizability of the research. The scope should also consider other healthcare professional such as physiotherapists, intensivists, respiratory therapists, and physicians to give the study team a broader insight into VAP prevention. This study relied on self-reported practices and self-administered questionnaires, which may have introduced bias, especially social desirability bias. This could explain the high rates of reported compliance despite the low knowledge scores. Therefore, more robust methods of data collection, for example, direct observation, are necessary to augment the accuracy of the data collected. Another limitation is that broader contextual influences, including workplace culture, the availability of clinical protocols, and team dynamics, were not assessed, even though they likely affect how nurses apply VAP prevention measures. Besides, a lack of specific and then validated tools to investigate VAP prevention knowledge and practices among CCNs might have consequences in the rehabilitation of outcomes by way of reliability of results, and emphasis on the needs for future research in development and validation of such tools. Moreover, the present study did take into consideration the direct associations between VAP prevention practices and patient outcomes, and the level of commitment from hospital leadership was considered as an essential element that guarantees the success of all attempts to prevent VAP. Future research needs to fill these gaps by developing better measurement tools, widening the context in which VAP prevention is taking place, and increasing our knowledge about CCNs' knowledge and compliance with practices within Malaysian healthcare settings.

## Conclusion

This study has indicated a gap in knowledge about the prevention of VAP, despite the reported high self-reported practice score by critical care nurses in Sarawak, Malaysia. Shortage of staff was identified as the major barrier to effective prevention of VAP. Nurses handle the care of mechanically ventilated patients, therefore the identification of such knowledge gaps and barriers is crucial if any improvement in patient safety and health outcomes is to be achieved. Future studies should be conducted on the factors affecting long-term knowledge retention, as well as behavioural and psychological factors such as stress and workload. Besides, the impact of newer guidelines on the incidence of VAPs and length of stay in an ICU has to be reviewed. Qualitative studies are also recommended to raise a more detailed description of problems considered by CCNs; better survey tools and overall review in all sectors of healthcare in Malaysia is necessary. The efforts below can support policy changes, address staffing concerns, and promote better training programs.

## Supporting information

**S1 Table. Item analysis of knowledge towards ventilator-associated pneumonia (VAP) prevention among critical care nurses.**
(DOCX)

**S2 Table. Item analysis of perceived practice towards ventilator-associated pneumonia (VAP) prevention among critical care nurses.**
(DOCX)

**S3 Table. Item analysis barriers towards ventilator-associated pneumonia (VAP) prevention among critical care nurses.**
(DOCX)

## Acknowledgments

We would like to thank the Director General of Health Malaysia for his permission to publish this article.

## Author contributions

**Conceptualization:** Jia Hui Tan.

**Formal analysis:** Jia Hui Tan.

**Investigation:** Jia Hui Tan.

**Methodology:** Jia Hui Tan.

**Project administration:** Jia Hui Tan.

**Resources:** Jia Hui Tan.

**Software:** Jia Hui Tan.

**Supervision:** Chong Chin Che, Li Yoong Tang, Mei Chan Chong.

**Validation:** Chong Chin Che.

**Writing – original draft:** Jia Hui Tan.

**Writing – review & editing:** Chong Chin Che.

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
