## [Decision Letter · Decision Letter 0]

11 Aug 2025

Dear Dr. Tan,

Thank you for submitting your manuscript to PLOS ONE. After careful consideration, we feel that it has merit but does not fully meet PLOS ONE’s publication criteria as it currently stands. Therefore, we invite you to submit a revised version of the manuscript that addresses the points raised during the review process.

**ACADEMIC EDITOR: Please see the comments to authors**  

We look forward to receiving your revised manuscript.

Kind regards,

Vijay Hadda, MD

Academic Editor

PLOS ONE

Journal Requirements:

Reviewers' comments:

Reviewer's Responses to Questions

**Comments to the Author**

1. Is the manuscript technically sound, and do the data support the conclusions?

Reviewer #1: Partly

Reviewer #2: Partly

2. Has the statistical analysis been performed appropriately and rigorously?

Reviewer #1: Yes

Reviewer #2: No

3. Have the authors made all data underlying the findings in their manuscript fully available?

Reviewer #1: Yes

Reviewer #2: Yes

4. Is the manuscript presented in an intelligible fashion and written in standard English?

Reviewer #1: Yes

Reviewer #2: Yes

Reviewer #1: The manuscript addresses an important and underexplored area—critical care nurses’ knowledge and practices regarding VAP prevention in a Malaysian context.

MAJOR CONCERNS

The term "perceived practice" is used throughout the manuscript but is conceptually vague. Instead self reported practice may be used

Self report introduces bias (social desirability bias) which is not described in the discussion part. This may be responsible for the high compliance reported despite poor knowledge.

Variables such as workplace culture, availability of clinical protocols, and team dynamics are not explored but likely influence practice, can be included in the discussion of limitations

Minor concerns- youtube link may not be appropriate as reference

(Faculty of Medicine University Malaya. Antimicrobial Resistance: Combating the silent 443 pandemic mini-symposium [Internet]. YouTube; 2022 Sep 10 [cited 2024 Apr 10]. 444 Available from: https://www.youtube.com/watch?v=HXF9mh6O2TQ. —- YOUTUBE LINK AS REFERENCE)

Reviewer #2: Dear authors,

I read the manuscript with keen interest. The muscript describes the knowledge and practice of nurses regarding VAP prevention bundle. The subject has been addressed previous a few times; however, it is still relevant. The manuscript has several limitations that needs to answered to make it better. Please find below my comments for the manuscript:

Data collection:

Authors mentioned “Responses with incomplete data or entry errors were excluded.” What was the threshold used? Any data missing or specific/critical data? If so, what are the critical data that lead to exclusion.

Knowledge of VAP prevention strategies

The total scale had a Cronbach's α of .64, the CVI was .89, and this scale was valid and reliable. Please spell out CVI.

Results

What was the response to questionnaire? How many nurses were invited to participate and how many responded? Response rate is important for the point of view of reproducibility and reliability of results.

Line 192-193 - Most participants were clinical staff nurses (74.8%) with 58.1% working fewer than 45 hours per week, and 98.0% were permanent staff members. The statement is confusing. I assume that all participants in this study were nurses. Then, what is meaning of “Most participants were clinical staff nurses (74.8%)”? Please clarify or rephrase the sentence.

Table 1.

What is “Post basic ICU’? Please use a terminology which is self-explanatory. Also, use full name for Pead, Neuro, ID, SN…

Table 2.

Table has classified participants in to Good, Acceptable and Poor Knowledge. However, the basis of classification is not clear from the manuscript.

Table 3

The title and variable for practice levels are confusing. It is not clear from the text what exactly is this table depicting. Does it depict – Perceived compliance to VAP preventive strategies Or is it actual compliance to the VAP Preventive strategies? Also, in table variable compliance and knowledge both are mentioned, which variable is correct?

Table 4

It is not clear. The table does not present any relationship/correlation (e.g. co-relation coefficient). Please modify the table. Also, table is usually used to present multiple variables/outcomes together for easy understanding. However, in this table only 1 variables are there. So, please think of alternative way of presentation.

Predictors of critical care nurses’ practice in VAP prevention.

Authors have described model 1 and model 2. However, the basis of modelling is not described in the methodology.

References

References should be preferable from Journals. However, there are many citations from the internet source. For example – ref 22. There are many trials on this subject. May cite the latest systematic review/metanalysis. Other references - 31

**Do you want your identity to be public for this peer review?** For information about this choice, including consent withdrawal, please see our Privacy Policy

Reviewer #1: No

Reviewer #2: No

---

## [Author Response · Author response to Decision Letter 1]

3 Sep 2025

We sincerely thank the reviewers for their valuable feedback.

I have carefully revised the manuscript and addressed all the comments provided. Specific actions were taken to improve the clarity, accuracy, and overall quality of the paper in line with the suggestions.

I truly appreciate the constructive input that has strengthened this work, and I respectfully submit the revised version for further consideration.

---

## [Decision Letter · Decision Letter 1]

11 Nov 2025

Dear Dr. Tan,

Thank you for submitting your manuscript to PLOS ONE. After careful consideration, we feel that it has merit but does not fully meet PLOS ONE’s publication criteria as it currently stands. Therefore, we invite you to submit a revised version of the manuscript that addresses the points raised during the review process.

Please edit the tables as suggested. 

We look forward to receiving your revised manuscript.

Kind regards,

Vijay Hadda, MD

Academic Editor

PLOS ONE

Journal Requirements:

Reviewers' comments:

Reviewer's Responses to Questions

**Comments to the Author**

Reviewer #1: All comments have been addressed

Reviewer #2: All comments have been addressed

2. Is the manuscript technically sound, and do the data support the conclusions?

Reviewer #1: Yes

Reviewer #2: Yes

3. Has the statistical analysis been performed appropriately and rigorously?

Reviewer #1: Yes

Reviewer #2: Yes

4. Have the authors made all data underlying the findings in their manuscript fully available?

Reviewer #1: Yes

Reviewer #2: Yes

5. Is the manuscript presented in an intelligible fashion and written in standard English?

Reviewer #1: Yes

Reviewer #2: Yes

Reviewer #1: (No Response)

Reviewer #2: Dear authors,

Thank you for responding to my comments. The manuscript has improved significantly. However, I have a minor comment:

-Please combine table 3 and 4. It will be better to presentation of the data in single table.

**Do you want your identity to be public for this peer review?** For information about this choice, including consent withdrawal, please see our Privacy Policy

Reviewer #1: **Yes: ** Sryma PB

Reviewer #2: No

---

## [Author Response · Author response to Decision Letter 2]

13 Nov 2025

We thank the reviewer for this helpful suggestion.

In response, Tables 3 and 4 have been merged into a single, integrated table titled “Table 3. Critical care nurses’ self-reported practice levels and correlation with knowledge on ventilator-associated pneumonia (VAP) prevention (N=298).”

The corresponding in-text references have been revised accordingly.

This update improves the clarity and presentation of the data, and no changes were made to the results or their interpretation.

---

## [Editor Report · Decision Letter 2]

18 Nov 2025

Predictors of Self-Reported Practice in Ventilator-Associated Pneumonia (VAP) Prevention Among Critical Care Nurses in Sarawak Public Hospitals

PONE-D-25-25193R2

Dear Dr. Tan,

We’re pleased to inform you that your manuscript has been judged scientifically suitable for publication and will be formally accepted for publication once it meets all outstanding technical requirements.

Kind regards,

Vijay Hadda, MD

Academic Editor

PLOS ONE

---

## [Editor Report · Acceptance letter]

PONE-D-25-25193R2

PLOS ONE

Dear Dr. Tan,

I'm pleased to inform you that your manuscript has been deemed suitable for publication in PLOS ONE. Congratulations! Your manuscript is now being handed over to our production team.

Kind regards,

on behalf of

Dr. Vijay Hadda

Academic Editor

PLOS ONE